# Constituents and Selective BuChE Inhibitory Activity of the Essential Oil from *Hypericum aciculare* Kunth

**DOI:** 10.3390/plants12142621

**Published:** 2023-07-12

**Authors:** James Calva, Carlos Ludeña, Nicole Bec, Christian Larroque, Melissa Salinas, Giovanni Vidari, Chabaco Armijos

**Affiliations:** 1Departamento de Química, Universidad Técnica Particular de Loja (UTPL), Loja 1101608, Ecuador; jwcalva@utpl.edu.ec (J.C.); ciludena@utpl.edu.ec (C.L.); masalinas4@utpl.edu.ec (M.S.); 2Institute for Regenerative Medicine and Biotherapy (IRMB), Université de Montpellier, National Institute of Health, and Medical Research (INSERM), 34295 Montpellier, France; nicole.bec@inserm.fr (N.B.); cjlarroque@gmail.com (C.L.); 3Department Nephrol Dialysis & Transplantat, Montpellier University Hospital, 34295 Montpellier, France; 4Department of Medical Analysis, Faculty of Applied Science, Tishk International University, Kurdistan Region, Erbil 44001, Iraq; vidari@unipv.it

**Keywords:** *Hypericum aciculare*, essential oil, constituents, BuChE selective inhibitory activity

## Abstract

A potential source of new inhibitors of cholinesterase enzymes are certain compounds of natural plant origin; therefore, in the study described herein we have determined the chemical composition and the acetylcholinesterase (AChE) and butyrylcholinesterase (BuChE) inhibitory activities of the essential oil (EO) steam distilled from aerial parts of *Hypericum aciculare*, which was collected in southern Ecuador. The oil qualitative and quantitative composition was determined by GC-FID and GC-MS using a non-polar and a polar chromatographic column. A total of fifty-three constituents were identified, that accounted for about 98% of the EO content. The hydrocarbon *n*-nonane (16.4–28.7%) and the aldehyde *n*-decanal (20.7–23.1%) were the predominant oil constituents. In addition, the EO showed significant inhibition of BuChE (IC_50_ = 28.3 ± 2.7 μg/mL) and moderate activity towards AChE (IC_50_ = 82.1 ± 12.1 µg/mL). Thus, the EO from *H. aciculare* aerial parts is an interesting candidate to investigate the mechanism of selective ChE inhibition by the two ChE enzymes with the aim to discover potential targets to control the progression of the Alzheimer’s disease (AD).

## 1. Introduction

The genus *Hypericum* (Hypericaceae) comprises about 500 species of herbs, shrubs, and trees, which are mainly distributed in temperate regions and on tropical mountains. They grow abundantly in Europe, Asia, Northern Africa, and America [1,2,3]. Several *Hypericum* taxa have a long traditional value as medicinal plants to treat inflammation, bacterial and viral infections, burns, stomach ulcers, mild depression, wounds and burns, diarrhea, pain, fevers, and poisoning by venomous animal bites [4,5]. Several biologically active secondary metabolites from *Hypericum* species exhibit a wide range of beneficial pharmacological effects, such as antimicrobial, anti-inflammatory, antioxidant, antifungal, astringent, antihyperglycemic, and hepatoprotective properties, as well as acetylcholinesterase and monoamine oxidase inhibitory activities [5,6]. Secondary metabolites isolated from *Hypericum* genus include naphthodianthrones (hypericin and pseudohypericin), ploroglucinols (hyperforin), flavonoids (quercetin, rutin, quercitrin, isoquercetin, hyperoside and amentoflavone), and essential oils [5], with antifungal [7], antibiotic [8], antiviral [9], and anticancer properties [10]. 

*Hypericum* essential oils (EOs), extracts, and isolated metabolites are reported to possess potent antidepressant, antibacterial, antifungal, antioxidant, anti-angiogenic, antiviral, antimalarial, cytotoxic, neuroprotective, tyrosinase inhibition, immunomodulatory, hepatoprotective, anti-inflammatory and wound-healing effects [11,12,13]. Thanks to these properties, *Hypericum* plants and products are widely used in traditional medicines practiced worldwide.

*Hyperycum aciculare* Kunth, commonly known in Perú by the indigenous names of “*Hierba de las cordilleras*”, “*Lechuguilla*”, “*Hierba de iman*” [14] and in Ecuador as “*Bura*” [15], is a 0.3–2 m tall shrub, erect, and bushy to decumbent and slender, with branches strict to ascending, lateral, and frequently pseudo-dichotomous [16]. An infusion of the plant is used by indigenous peoples who live in the Carchi province of Ecuador to treat wounds and furuncles of animals [17]. Moreover, to relieve rheumatic pains, the flowers of *H. aciculare* are crushed, mixed with animal fat, and placed on the affected part. The analgesic effect of this preparation is enhanced by adding other species of *Hypericum*, such as *H. decandrum*, that has the same therapeutic properties. *H. aciculare* is also used by different indigenous Andean communities as firewood for cooking food, to combat bad luck, and as a protector against evil spirits [15].

*H. aciculare* is native to the Andes. In southern Ecuador it is distributed in the provinces of Azuay and Loja, at altitudes of 1500–4000 m a.s.l, while in northeastern Ecuador, it mainly grows in more humid environments, on slopes where the bamboo also occurs [18]. *H. aciculare* has also been reported to grow on the Cordillera Oriental of Columbia as an indicator of biodegradable or polluted soils due to livestock overgrazing, surface erosion, agrochemical pollution, and frequent fires [16].

This research is part of our ongoing program on the study and valorization of essential oils from Ecuadorian aromatic plants with potential acetylcholinesterase inhibitory effects [19]. In continuation of our studies on the essential oils from species of the genus *Hypericum* [20], we examined the chemical composition and the activity of the EO steam distilled from H. aciculare as a potential cholinesterase inhibitor. There are two important cholinesterase enzymes in the human body: acetylcholinesterase (AChE), which is found in red blood cells as well as in the lungs, spleen, nerve endings, and the gray matter of the brain, and butyrylcholinesterase (BuChE), which occurs in the serum as well as in the liver, muscle, pancreas, heart, and white matter of the brain [21]. Acetylcholinesterase is involved in the transmission of nerve impulses by breaking down acetylcholine, a molecule that helps transmission of the signals across nerve endings. Recent evidence suggests that both enzyme (AChE and BuChE) may have roles in the etiology and progression of Alzheimer’s disease (AD) in addition to the regulation of synaptic ChE levels in healthy brain. In fact, decrease in brain acetylcholine (ACh) levels is implicated in the pathophysiology of cognitive dysfunction occurring in Alzheimer’s disease (AD). Therefore, the inhibition of the ACh catabolic enzymes, AChE and butyrylcholinesterase BuChE, can contribute to increase ACh brain levels. Moreover, it has been hypothesized that inhibition of AChE and BuChE may contribute to counter formation Aβ plaques and therefore to represent a disease-modifying strategy principle [22]. These features make acetylcholinesterase inhibitors (AChEi) and butyrylcholinesterase inhibitors (BuChEi) the main class of drugs currently used for the treatment of the AD dementia phase. Moreover, studies have established the therapeutic interest of the inhibition of both AChE and BuChE [22]. In fact, BuChE activity progressively increases in patients with AD, while AChE activity remains unchanged or declines in some brain areas where the BuChE:AChE ratio may shift from 0.6 to as high as 11. 

The therapeutic potential of natural products against Alzheimer’s disease has been highly recognized [23,24]. However, the alkaloid galanthamine is the only naturally occurring substance that is in the market for the therapy of mild or moderate forms of Alzheimer’s disease, and other disorders of memory [22]. Recently, essential oils (EOs) isolated from medicinal and aromatic plants are receiving an increasing attention as potential neuroprotective remedies for age-related neurodegenerative diseases, due to their cholinesterase inhibitory activity against AChE and BuChE [25]. 

Herein, we describe the still unknown chemical composition and the inhibitory activity against AChE and BuChE of the EO obtained by steam distillation from *H. aciculare*.

## 2. Results

### 2.1. Physical Properties

Steam distillation of fresh leaves, flowers, and stems of *H. aciculare* produced a yellowish EO with a yield of 0.038 ± 0.03 % (*w*/*w*) on three replicates. The physical properties of the oil were: density = 0.88 ± 0.02 g/mL; refractive index = 1.46 ± 0.04; the specific rotation = + 0.54 ± 0.01 g/mL. 

### 2.2. Chemical Composition 

Fifty-three constituents were determined in the *H. aciculare* EO by GC-FID and GC-EIMS analysis, that represented 98.02% and 98.43% of the total oil analyzed on a DB-5ms and a HP-INNOWax column, respectively. The major constituents were alkanes and aldehydes, accounting for 20.97–31.62% and 21.68–23.82% of the total EO, respectively. The terpenoid fraction included sesquiterpene hydrocarbons (17.19, 21.22%), oxygenated sesquiterpenoids (3.35, 5.44%), oxygenated monoterpenoids (10.21, 10.65%), and monoterpene hydrocarbons (5.09, 7.98%).

The main identified compounds were *n*-nonane and *n*-decanal with a percent content of 28.71 and 16.36%, and 20.68 and 23.12%, respectively, in the EO analyzed on a DB-5ms capillary column and a HP-INNOWax capillary column, respectively.

The mean percent content of each identified component and the standard deviation (SD) were calculated, over three replicates, from the corresponding, electronically integrated, peak area relative to the total area of the peaks in each GC-FID chromatogram. No correction factor was applied. The chemical composition of the EO from *H. aciculare* is reported in Table 1, whereas the gas chromatograms on the two columns are shown in Figure 1 and Figure 2.

### 2.3. Inhibition of Cholinesterases 

The anti-cholinesterase activity of *H. aciculare* EO has been evaluated for the first time. The oil exhibited moderate inhibitory activity against AChE, with an IC_50_ value of 82.1 ± 12.1 µg/mL, and an interesting activity against BuChE, with an IC_50_ = 28.3 ± 2.7 μg/mL (Figure 3). The synthetic drug donepezil hydrochloride, which is employed to treat mild-moderate forms of Alzheimer’s dementia, was used as the positive control in the test. It showed IC_50_ (μg/mL) values of 0.04± 0.02 and 3.60 ± 0.20 against AChE and BuChE, respectively [20]. 

To discard the possibility that the loss of enzymatic activity could be due to protein denaturation by the EO from *H. aciculare*, we determined the activity of the enzyme horseradish peroxidase (HRP) towards the substrate 2,2′-azino-di-[3-ethylbenzthiazoline-6-sulfonic acid] (ABTS), in the presence of the oil, which was compared to the enzymatic activity in the sole vehicle (DMSO). As clearly indicated by the curves reported in Figure 4, the enzymatic activity was not reduced in the presence of the oil at the high concentration used, thus attesting that the oil do not cause protein denaturation. 

## 3. Discussion

The average yield of the EO steam distilled from *H. aciculare* aerial parts was 0.038% (*w/w*), confirming that *Hypericum* is generally an EO-poor genus, with yields <1% *w*/*w* [11,13] although in a few cases higher EO yields, up to 3%, have been reported in the literature [13]. The *H. aciculare* EO yield was even lower than those of other *Hypericum* EOs, such as *H. silenoides* (0.1–0.6%) and *H. philonotis* (0.2%), collected in central Mexico [32], or *H. laricifolium* (0.15%) collected in southern Ecuador [19]. The mean density and refractive index values were like those reported for the *H. laricifolium* EO, while the signs of the specific rotation of the two oils were opposite [19].

Yield, content, and chemical composition of *Hypericum* EOs show extensive inter- and intraspecific variation [12,13], which may be related to the influence of variables such as genetic factors, developmental stage and phenological cycle of seasonal changes, type of plant material and other conditions, such as geographical distribution [13]. An example of such variability is the significant qualitative and quantitative differences reported for the chemical profiles of the EOs from *H. perforatum* [11,13]. 

Regarding the chemical composition, *Hypericum* EOs can roughly be divided in three groups: (i) those in which terpenoids prevail; (ii) those in which non-terpenoid aliphatic derivatives are predominant; (iii) those containing comparable amounts of both types of compounds [11,12,13]. The typical most abundant terpenoids include the monoterpenes α- and β-pinenes, limonene, and β-ocimene, and the sesquiterpenes germacrene-D, cadinenes, selinenes, β-caryophyllene, and caryophyllene oxide, whereas linear C_9_-C_12_, C_16_, C_25_, C_29_, and C_30_ hydrocarbons are among the most frequently found aliphatic compounds [11,12,13]. The chemical composition of essential oils from the genus *Hypericum* has been used as chemotaxonomic marker to differentiate taxa of this genus. However, such a hypothesis is no longer accepted, after that more data from a wide range of geographic distributions, taxonomic ranks, and seasonality have been examined [13]. 

Terpenoid constituents, particularly sesquiterpenes, were described as primary constituents in the oil of *Hypericum* collected from mountainous regions, while long-chain waxes and fats were reported as dominating in plants from lowland areas [11]. It was suggested that the increasing concentrations of monoterpenoids and sesquiterpenoids with increasing altitudes, was due to the role of these compounds in helping the plant deal with abiotic stress factors, e.g., UV radiation. Our findings, however, contradicted this hypothesis. In fact, the EO of *H. aciculare*, that was collected at an altitude of 3000 m, was significantly rich in non-terpenoid derivatives, among which *n*-nonane and *n*-decanal were predominant. However, their estimated percent content in the oil significantly depended on the type of the gas chromatographic column used for the analysis (Table 1). For example, the amount of *n*-nonane was determined as 28.71% and 16.36%, on a DB-5ms capillary column and a HP-INNOWax capillary column, respectively. Other *Hypericum* species producing *n*-nonane as the main EO component were *H. philonotis* Schltdl. & Cham (35.9%) and *H. silenoides* Juss. (31.9%) from Contepec-Mexico [32], *H. grandifolium* Choisy (42.3%) [33], *H. canariense* L. (44.3%) [33], *H. patulum* Thunb (17.1–32.6%) [34], *H. rochelii* Griseb. & Schenk (24.7%) [35], and *H. japonicum* Thunb. ex Murray (21.4%) [36]. Interestingly, EOs from *H. silenoides* [32] and *H. japonicum* [36] also contained significant amounts of *n*-decanal, 15.2 and 8.2%, respectively, thus resembling the content of *H. aciculare*. 

Bioactivities of *Hypericum* EOs most frequently tested until recently are those traditionally investigated with essential oils, i.e., antimicrobial, insecticidal, antioxidant, antiviral, and cytotoxic properties [11,12,13]. In recent studies, other properties, such as neuroprotective, tyrosinase inhibitory, immunomodulatory, antiangiogenetic, hepatoprotective, and wound-healing effects, have been assayed. However, little is known about the cholinesterase inhibitory activity of *Hypericum* EOs, despite the increasing interest in essential oils as potential neuroprotective remedies for age-related neurodegenerative diseases [23]. In pioneering studies, the EO (0.5 mg/mL) of *H. undulatum* showed moderate inhibitory activity (20.0 ± 6.5 %) against AChE, which was much lower than that of the plant extract whose inhibitory activity was 68.4 ± 4.7% at a concentration of 0.5 mg/mL [37]. At higher concentration (10 mg/mL), *H. neurocalycinum* and *H. malatyanum* inhibited 85.78 ± 4.11 and 62.24 ± 1.81%, respectively, of the AChE activity [38]. 

In our studies of essential oils with anticholinesterase properties, we were the first to find selective inhibition of ChE enzymes by *Hypericum* EOs. In fact, the EO of *H. laricifolium* exhibited an IC_50_ of 36.80 ± 2.4 μg/mL against BuChE which was three times lower than against AChE (106.10 ± 20.20 μg/mL) [20]. The EO from *H. aciculare* displayed higher and more selective inhibition of ChE enzymes than the EO from *H. laricifolium*, with an IC_50_ = 28.3 ± 2.7 µg/mL against BuChE and an IC_50_ = 82.1 ± 12.1 µg/mL against AChE. This finding is highly interesting since BuChE has become an important new target in Alzheimer’s disease and related dementias, due to its importance in the later stages of AD [39,40]. In both enzymes, 55% of the amino acid sequences that form them are identical; however, due to substitution by aliphatic amino acids (6 of the 14 amino acids present) at the BuChE active site, the BuChE active site has a larger volume than the AChE active site.

Consequently, the BuChE active site is larger in volume than AChE, resulting in similar, however, distinct catalytic properties, substrate specificity, kinetics, and activity in different brain regions [39]. Studies on BuChE inhibition are not as numerous as for AChE and the relationship between activity and structures of inhibitors is far from having been firmly determined, although it has been observed that the efficacy of an agent to inhibit BuChE is determined by its ability to access binding sites at the base of the large gorge of BuChE [39]. Regarding the essential oils, which are complex mixtures of compounds with different chemical characteristics, the properties of bio-inhibition can be explained due to the individual effects of some of the main components of the oil as mono- and sesquiterpenes [19]. However, monoterpenes such as α -pinene and limonene, which showed relatively good inhibitory activity against equine serum BuChE [41], are minor components of the EO of *H. aciculare* (Table 1). Moreover, there is no information regarding specific anticholinesterase effects of the main oil constituents, *n*-nonane and *n*-decanal, that have structures and physicochemical characteristics significantly different from those of the compounds actively interacting with the binding sites of the ChE enzymes [42]. Instead, we assumed that these two compounds could have an indirect role in the modulation of the ChE activity, by facilitating the access of numerous minor components occurring in the EO (see Table 1) to the active site of the enzymes. For this reason, we didn’t engage ourselves in in silico studies of the major components of the EO from *H. aciculare*. Moreover, *in vivo* studies have demonstrated that exposure of rats to high concentrations of *n*-nonane can cause several cerebellar damages and a significant loss of Purkinje cells [43]. 

Therefore, it appears more appropriate to attribute the anticholinesterase activity of the EO of *H. aciculare* to an undefined interaction of the enzymes with different oil constituents, which may include synergistic and antagonistic relationships. This hypothesis has already been proposed as a mechanism of action of other types of essential oils [44,45]. Moreover, dedicated studies are needed to evaluate the biological activity of single compounds isolated from an EO, so to determine their potential contribution to the biological effects of the entire oil. An explanatory example of such studies is the isolation of the sesquiterpene acorenone B from the EO of the Ecuadorian plant *Niphogeton dissecta* (Benth.) J.F. MacBr, that, with an IC_50_ of 10.9 ug/mL, mainly contributed to the BuChE inhibitory activity of the oil (IC_50_ of 11.5 ug/mL). [46]. As an alternative to in vitro studies, computational approaches may be used to assess potential biological targets for the EO components [42,47].

## 4. Materials and Methods

### 4.1. Plant Material 

Aerial parts of *Hypericum aciculare* Kunth were collected on Loma Torre Mountain (Las Antenas sector), at the border between the Saraguro and San Lucas parishes, in the Loja province of Ecuador (Figure 5b), at an altitude of 3000 m above sea level (GPS coordinates: −3.681639 N, −79.228698 E). The plant (Figure 5a) was identified by Jose Miguel Andrade Morocho, botanist at the Universidad Técnica Particular de Loja (UTPL). A voucher has been deposited at the Herbarium of UTPL under the accession code HUTPL5576. The plant collection was authorized by a governmental permission (MAE-DBN-2016-048).

### 4.2. Distillation of the Essential Oil 

The fresh plant material (aerial parts) was steam distilled in a Clevenger apparatus for 3 h in three replicates of approximately 2000 g each, the three distillates were pooled, and the pale yellow EO (approximately 0.75 g from each distillation) was dried with anhydrous Na_2_SO_4_ and stored in brown vials at −4°C until biological and chemical analyses [14].

### 4.3. Physical Properties 

Three physical properties were determined for the EO: the relative density, according to standard AFNOR NF T75-111; the refractive index according to standard ANFOR NF 75-112, using an ABBE refractometer (Germany); the specific rotation [∝]_D_, according to ISO 592-1998 using an automatic polarimeter Hanon P 810. For each physical property, the mean value and the SD were measured from tree replicates performed at 20 °C.

### 4.4. Chemical Composition

According to standard procedures, the EO chromatographic analysis was performed by gas chromatography and gas chromatography-mass spectrometry (GC-MS) techniques, while a flame ionization detector (GC-FID) was used for quantification of the EO components. Each EO component was identified by comparing the mass spectrum and the calculated linear retention index (LRI) with the corresponding data reported in the literature. The LRIs were calculated according to the method of Van Den Dool and Kratz [48], using a mixture of *n*-alkanes (C9-C25) injected under the same chromatographic conditions as the EO samples. 

#### 4.4.1. GM-MS Analysis

A Thermo Scientific Gas Chromatographer (Trace 1310) with electronic pressure control was used for the gas chromatography (GC) analysis, this apparatus was coupled to a simple quadrupole mass spectrometry detector ISQ 7000 (Thermo Fisher Scientific, Wal-than, MA, USA). Two columns were used: a DB-5ms column (5% phenylmethylpolysiloxane, 30 m × 0.25 mm i.d., 0.25 μm film thickness) and a HP-INNOWax column (polyethylene glycol, 30 m × 0.25 mm i.d., 0.25 μm film thickness). The column temperature was programmed from an initial temperature of 60 °C for 5 min, followed by a gradient of 2 °C/min to 100 °C, followed by a gradient of 3 °C/min to 150 °C, followed by a gradient of 5 °C/min to 200 °C. Finally, the temperature was increased at 15 °C/min to 250 °C and held for 5 minutes. Ultrapure helium at a flow rate of 1 mL/min was used as a carrier gas and 1 μL of the EO solution in CH_2_Cl_2_ (1% *v/v*) was injected (split 40:1) [20,46,49]. 

#### 4.4.2. GC-FID Analysis

For the GC-FID analysis, the gas-chromatograph and the chromatographic operating conditions were the same as those described in the previous section. 

### 4.5. Cholinesterase Assays

The cholinesterase inhibitory activity of the *H. aciculare* EO was determined against acetylcholinesterase (AChE, from *Electrophorus electricus*, Sigma-Aldrich, C3389, St. Louis, MO, USA) and butyrylcholinesterase (BuChE, from equine serum, Sigma-Aldrich, SRE020, St Louis., MO, USA) [49]. AChE and BuChE inhibitory effects of the oil were measured in vitro in accordance with the procedure initially designed by Ellman [50], which we subsequently adapted to essential oils, as described in [46,49,51]. Measurements were made in triplicate. Donepezil was used as the reference ChE inhibitor. False positive results, due high concentration (>100 µg/mL) of amines or aldehydes, were not excluded [20,46,49]. Used as a control, the horseradish peroxidase activity versus its substrate, 2,2′-azino-di-(3-ethylbenzthiazoline-6-sulfonic acid) (ABTS), was determined using the increase in the absorbance at 405 nm (see Figure 4), in the equipment described for cholinesterase assays.

## 5. Conclusions

The physical properties, chemical composition, and anticholinesterase inhibitory properties of *Hypericum aciculare* EO were investigated and described for first time in this paper. The main component of the oil was the hydrocarbon *n*-nonane which contributed about 30% to the oil composition. The EO showed dose-dependent inhibition of the ChE activities, with an interesting selectivity against BuChE. We believe that these findings cast the basis of future studies aimed at determining the oil component(s) responsible for the inhibitory effects and at defining the kinetic mechanisms exerted against the cholinesterase enzymes. Our ultimate goal is to explore the possibility to include the *H. aciculare* EO among the natural therapeutic remedies for the treatment of AD and other dementia diseases.

## Figures and Tables

**Figure 1 plants-12-02621-f001:**
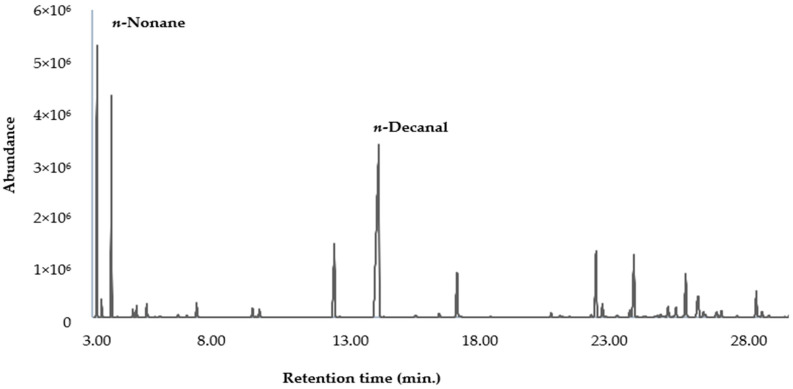
Gaschromatogram of the EO from *H. aciculare* aerial parts using a DB-5ms column.

**Figure 2 plants-12-02621-f002:**
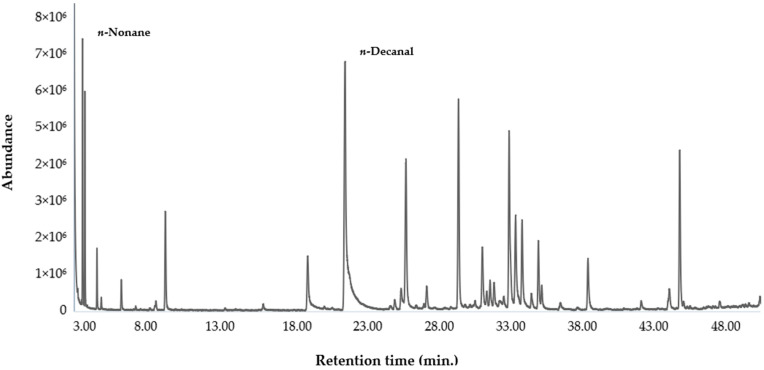
Gaschromatogram of the EO from *H. aciculare* aerial parts on a HP-INNOWax column.

**Figure 3 plants-12-02621-f003:**
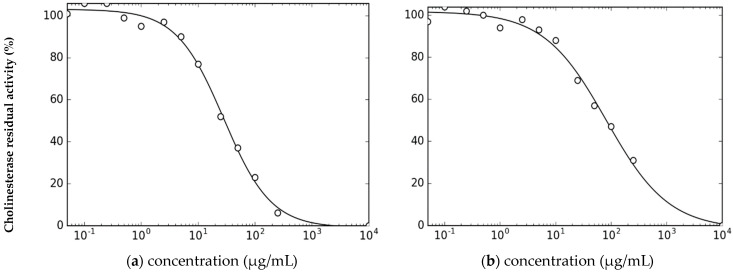
Curves of the cholinesterase residual activity (%) vs. the concentration (μg/mL) of the EO from *H. aciculare* aerial parts, used to calculate the IC_50_ values against (**a**) AChE and (**b**) BuChE.

**Figure 4 plants-12-02621-f004:**
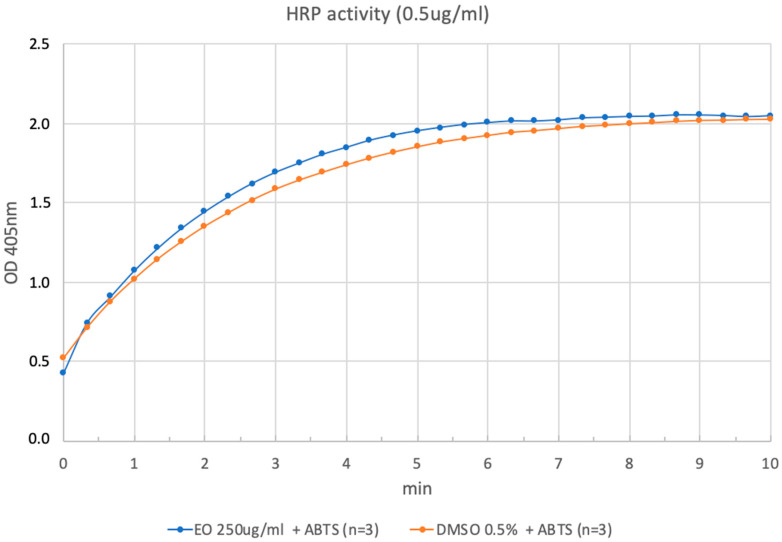
Curves of the horseradish peroxidase activity (HRP) vs. ABTS, in the presence of the EO from *H. aciculare* aerial parts, compared to the enzymatic activity in the sole vehicle (DMSO).

**Figure 5 plants-12-02621-f005:**
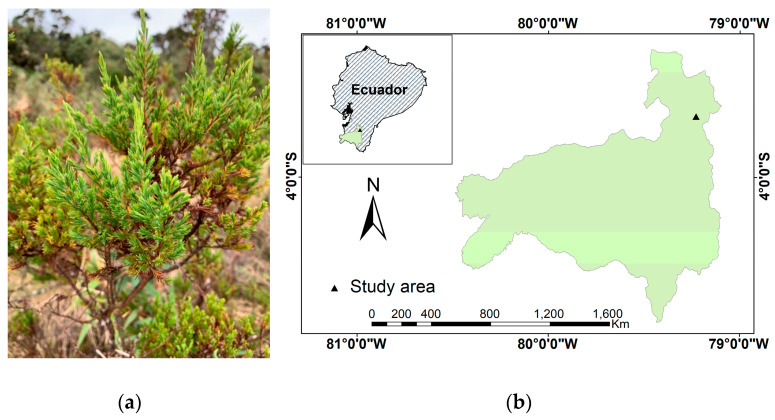
(**a**) Image of *H. aciculare* from Ecuador; (**b**) map of the collection site. Photographs obtained by one of the authors (C.A.).

**Table 1 plants-12-02621-t001:** Chemical composition of *Hypericum aciculare* essential oil.

		DB-5ms			HP-INNOWax			
N°	Compound	LRI ^a^	LRI ^b^	% ± SD	LRI ^a^	LRI ^c^	Ref.	% ± SD
1	*n*-Octane	800	800	0.20 ± 0.01	-	-	-	-
2	2-Methyl-octane	855	858	1.68 ± 0.02	860	858	[26]	3.26 ± 0.18
3	*n*-Nonane	900	900	28.71 ±0.61	904	900	[27]	16.36 ± 1.18
4	α-Pinene	930	932	2.05 ± 0.03	1046	1055	[25]	3.38 ± 0.38
5	Isoamyl propionate	968	960	0.29 ± 0.01	-	-	-	-
6	*β*-Pinene	975	974	0.17 ± 0.00	1109	1108	[25]	0.25 ± 0.02
7	Myrcene	988	988	0.69 ± 0.02	1161	1163	[25]	0.90 ± 0.07
8	*n*-Decane	1000	1000	0.02 ± 0.03	-	-	-	-
9	Limonene	1027	1024	0.10 ± 0.01	-	-	-	-
10	*(Z)-β*-Ocimene	1035	1032	0.15 ± 0.01	1236	1236	[25]	0.36 ± 0.02
11	(*E)-β*-Ocimene	1045	1044	1.93 ± 0.07	1254	1253	[25]	3.09 ± 0.26
12	*n*-Undecane	1100	1100	1.01 ± 0.00	1097	1100	[27]	1.35 ± 0.10
13	*n*-Nonanal	1105	1100	0.52 ± 0.03	1401	1395	[28]	0.36 ± 0.06
14	*(2E)*-Nonenol	1170	1163	4.02 ± 0.11	1457	-	-	3.85 ± 0.56
15	*n*-Decanal	1207	1201	20.68 ± 0.3	1504	1498	[28]	23.12 ± 2.73
16	Linalool	-	-	-	1566	1554	[25]	0.50 ± 0.02
17	Citronellol	1228	1223	0.53 ± 0.31	1783	1772	[25]	1.04 ± 0.06
18	1-Nonanol	-	-	-	1676	1664	[29]	0.48 ± 0.02
19	*(E)*-Ocimenone	1237	1235	3.30 ±0.06	-	-	-	-
20	*ϒ*-Selinene	-	-	-	1698	1690	[29]	0.71 ± 0.02
21	Linalyl acetate	1251	1254	1.35 ± 0.06	-	-	-	-
22	*(2E)*-Decenal	1268	1260	0.13 ± 0.09	-	-	-	-
23	Geranial	1268	1264	4.53 ± 0.06	1740	1732	[25]	5.32 ± 0.15
24	1-Decanol	1274	1266	1.05 ± 0.04	1778	1766	[30]	2.80 ± 0.19
25	Pearlate	1278	1287	3.23 ± 0.09	-	-	-	-
26	*2*-Undecanone	1292	1293	0.07 ± 0.05	1603	1604	[30]	0.22 ± 0.02
27	(2*E)*-Undecenal	1364	1357	0.09 ± 0.07	-	-	-	-
28	Cyclosativene	1372	1369	0.06 ± 0.09	-	-	-	-
29	Linalyl isobutanoate	1378	1373	0.49 ± 0.02	-	-	-	-
30	*β*-Longipinene	1399	1400	0.07 ± 0.00	-	-	-	-
31	*(E)-β*-Caryophyllene	1414	1417	4.59 ± 0.06	1580	1593	[28]	5.76 ± 0.22
32	Dodecanal	1409	1408	0.26 ± 0.01	1713	1715	[28]	0.34 ± 0.09
33	*α*-Guaiene	1437	1437	0.10 ± 0.02	-	-	-	-
34	*α*-Humulene	1450	1452	4.92 ± 0.04	1653	1658	[25]	6.62 ± 0.27
35	9-*epi-(E)-C*aryophyllene	1469	1464	0.36 ± 0.36	-	-	-	-
36	*γ*-Himachalene	1479	1481	0.01 ± 0.07	-	-	-	-
37	*β*-Selinene	1483	1489	0.81 ± 0.09	1704	1708	[28]	0.99 ± 0.04
38	*α*-Selinene	-	-	-	1734	1744	[26]	1.75 ± 0.06
39	*α*-Zingiberene	1490	1493	3.79 ± 0.05	-	-	-	-
40	*2*-Tridecanone	1494	1495	0.23 ± 0.00	-	-	-	-
41	*(E,E)-α*-Farnesene	1503	1505	2.47 ± 0.02	1751	1751	[25]	4.47 ± 0.16
42	*(Z)-γ*-Bisabolene	1507	1514	0.01 ± 0.00	1760	1741	[25]	0.92 ± 0.02
43	Geranyl acetate	-	-	-	1766	1761	[25]	0.73 ± 0.01
44	Nerol	-	-	-	1816	1805	[25]	0.39 ± 0.02
45	Geraniol	-	-	-	1867	1854	[25]	2.67 ± 0.15
46	Caryophyllene oxide	-	-	-	1971	1967	[25]	0.40 ± 0.01
47	Methyleugenol	-	-	-	2034	2028	[27]	0.44 ± 0.03
48	*(E)*-Nerolidol	1560	1561	2.89 ± 0.03	2060	2050	[19]	5.04 ± 0.38
49	*trans-β*-Elemenone	1595	1602	0.10 ± 0.09	-	-	-	-
50	Khusimone	1604	1604	0.26 ± 0.18	-	-	-	-
51	*γ*-Eudesmol	-	-	-	2175	2185	[27]	0.14 ± 0.06
52	Bisaboladien-4-ol	1618	1618	0.10 ± 0.07	-	-	-	-
53	Decanoic acid	-	-	-	2280	2284	[25]	0.42 ± 0.07
Aliphatics								
Alkanes (%)				31.62				20.97
Alcohols (%)				5.07				7.13
Aldehydes (%)				21.68				23.82
Ketones (%)				0.30				0.22
Fatty acids (%)				--				0.42
Esters (%)				3.52				--
Terpenes								
Monoterpene hydrocarbons (%)				5.09				7.98
Oxygenated monoterpenoids (%)				10.2				10.65
Sesquiterpene hydrocarbons (%)				17.19				21.22
Oxygenated sesquiterpenes (%)				3.35				5.44
Diterpenes (%)				-				0.14
Others (%)				-				0.44
TOTAL identified (%)				98.02				98.43

^a^ Calculated linear retention index; ^b^ linear retention index from reference [31]; ^c^ linear retention index from references [25,26,27,28,29,30]; %, mean percent content in the EO over three determinations; SD, mean standard deviation over three determinations.

## Data Availability

The original data can be requested from the authors (J.C.; M.S.; N.B. and C.L.).

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
