# Peer review of "Constituents and Selective BuChE Inhibitory Activity of the Essential Oil from Hypericum aciculare Kunth"

_plants, 2023, doi:10.3390/plants12142621_

Round 1
Reviewer 1 Report
The authors performed a study entitled Constituents and selective BuChE Inhibitory Activity of the essential oil from Hypericum aciculare Kunth"
In general, the manuscript has quality and can be accepted after revisions,
I recommend that figure 1 used in the introduction be moved to materials and methods.
In addition, to make the work richer, the authors should report more about essential oils as inhibitors, I recommend that the articles be cited.
https://doi.org/10.3390/molecules27144373
https://www.tandfonline.com/doi/full/10.1080/14786419.2021.1893724
https://doi.org/10.1021/jf804013j
https://doi.org/10.3390/ph10040084
In results, chromatograms can be sent for supplementary material
for the work to be richer, I recommend that the authors carry out an in silico study of the majority compounds, it can be a Docking using the AChE enzyme and BuChE as a molecular target.
In conclusion, the authors should show the advances made in the present work and not focus on data already discussed.
Author Response
Reviewer 1:
The authors performed a study entitled Constituents and selective BuChE Inhibitory Activity of the essential oil from Hypericum aciculare Kunth"
In general, the manuscript has quality and can be accepted after revisions,
Dear reviewer, thank you for your comments and recommendations to improve our article. Changes have been highlighted in blue in the manuscript.
-I recommend that figure 1 used in the introduction be moved to materials and methods.
This change has been made. Now (figure 4) is in the materials and methods section.
-In addition, to make the work richer, the authors should report more about essential oils as inhibitors, I recommend that the articles be cited.
https://doi.org/10.3390/molecules27144373
https://www.tandfonline.com/doi/full/10.1080/14786419.2021.1893724
https://doi.org/10.1021/jf804013j
https://doi.org/10.3390/ph10040084
We have added the suggested citations. See lines 231-239, cites [44-47]
-In results, chromatograms can be sent for supplementary material
Now chromatograms (figures 1 and 2) are attached as supplementary material.
-for the work to be richer, I recommend that the authors carry out an in silico study of the majority compounds, it can be a Docking using the AChE enzyme and BuChE as a molecular target.
We thank the referee for this advice, it is noticeable that an in-silico study would enrich the article. A recent revue review, using in silico approach, review the close interaction between natural compounds and active site of AChE and/or BuChE [42]. The two major compounds, n-nonane and n-decanal, identified from the EO, doesn’t have a structure close to that described in this review. We guess that they could facilitate the access to numerous minor compounds identified (see table 1) to the active site of the enzymes, so we didn’t engage structural studies with these compounds. The reference of this paper and a sentence is now added in the text. See lines 225-226
In conclusion, the authors should show the advances made in the present work and not focus on data already discussed.
Suggested changes have been made in the new manuscript.

Reviewer 2 Report
Accept
Author Response
Reviewer 2
There are no suggested changes.

Reviewer 3 Report
1. In Line 114, the sentence exhibits a significantly large range of 8.05-31.62%. Please provide an explanation for the observed magnitude of this range.
2. The manuscript contains inconsistent column names, "DB-5MS" and "HPINNOWax". Please make the necessary revisions to ensure consistency.
3. In Figure 3, the numbers on the x-axis have been represented with commas, whereas they should be depicted with decimal points. Please make the appropriate revision.
4. Figure 4 displays IC50 values that differ from those mentioned in the manuscript. Kindly verify if the calculated values are accurate.
5. There is a significant concern regarding the enzyme activity inhibition. It should be noted that enzymes (proteins) have the potential to undergo denaturation in the presence of oil. Therefore, it is essential to consider whether the essential oil derived from Hypericum aciculare may denature the enzymes instead of inhibiting their activity. In such a case, the observed effect would not be classified as inhibitory activity.
6. In line 189, the values 3.44 and 28.71% appear to be contradictory. Please revise the provided information accordingly.
Author Response
Reviewer 3
Thank you for your comments and recommendations to improve our article Changes made on the manuscript are highlighted in gray.
- In Line 114, the sentence exhibits a significantly large range of 8.05-31.62%. Please provide an explanation for the observed magnitude of this range.
Thank you for your observation, there was an identification error in the polar column (HP-INNOwax)-FID, the authors discarded a peak corresponding to the compound n-nonane and confused it with the solvent, the corresponding correction was made in line 115, as in Table 1, the correct values are (20.97 and 31.62%).
- The manuscript contains inconsistent column names, "DB-5MS" and "HPINNOWax". Please make the necessary revisions to ensure consistency.
Done, we have made the suggested changes.
- In Figure 3, the numbers on the x-axis have been represented with commas, whereas they should be depicted with decimal points. Please make the appropriate revision.
Thank you, we have made the suggested changes.
- Figure 4 displays IC50 values that differ from those mentioned in the manuscript. Kindly verify if the calculated values are accurate.
In figure 4 (now 3) is clearly between 10 and 100 µg/ml as reported in the text: AChE IC50 = 82.1 ± 12.1 µg/mL (left panel) and BuChE, IC50 = 28.3 ± 2.7 µg/mL (right panel). The IC50 determination for Donepezil is not represented.
See line 136
- There is a significant concern regarding the enzyme activity inhibition. It should be noted that enzymes (proteins) have the potential to undergo denaturation in the presence of oil. Therefore, it is essential to consider whether the essential oil derived from Hypericum acicularemay denature the enzymes instead of inhibiting their activity. In such a case, the observed effect would not be classified as inhibitory activity.
We agreed with the referee that the loss of enzyme activity could be the consequence of protein denaturation as well as effective inhibition of the activity. To evaluate this possibility, we study the horseradish peroxidase activity in presence of the highest concentration of the EO vs vehicle only (DMSO). As presented under, EO from Hypericum aciculare have no effect on HRP activity. This experiment was reported in the text as data not shown.
See lines 306-309
- In line 189, the values 3.44 and 28.71% appear to be contradictory. Please revise the provided information accordingly.
Thank you, the correct values are 16.36 and 28.71% after correcting the ID, (line 181).

Round 2
Reviewer 1 Report
The authors carried out the suggested revisions, I recommend publication of the manuscript
Reviewer 3 Report
None